# Exact solutions of (1 + 1)-dimensional integro-differential Ito, KP hierarchy, CBS, MCBS and modified KdV-CBS equations

**Amina Amin** [1]*, **Imran Naeem**[2], **Adnan Khan**[1]

**1** Department of Mathematics, National College of Business Administration & Economics, Lahore Gulberg III, Pakistan, **2** Department of Mathematics, Lahore University of Management Sciences, Lahore Cantt., Pakistan

* 82083326@ncbae.edu.pk, amina_zee@hotmail.com

## Abstract

The present study computes the Lie symmetries and exact solutions of some problems modeled by nonlinear partial differential equations. The (1 + 1)-dimensional integro-differential Ito, the first integro-differential KP hierarchy, the Calogero-Bogoyavlenskii-Schiff (CBS), the modified Calogero-Bogoyavlenskii-Schiff (CBS), and the modified KdV-CBS equations are some of the problems for which we want to find new exact solutions. We employ similarity variables to reduce the number of independent variables and inverse similarity transformations to obtain exact solutions to the equations under consideration. The sine-cosine method is then utilized to determine the exact solutions.

## 1 Introduction

Nonlinear partial differential equations (NLPDEs) have been utilized to characterize various nonlinear occurrences in mathematical biology, physics, and several other areas of science and engineering. The most important problem in real-world phenomena is computing exact solutions of nonlinear PDEs. The homogeneous balance method [1], the Darboux transform method [2, 3], the first integral method [4, 5], the tanh function method [6], the modified simple equation method [7, 8], the method of the auxiliary equation [9, 10], the $(G'/G)$-expansion method [11, 12], the F-expansion method [13, 14], Jacobi elliptic function method [15, 16], and Lie symmetry method [17–19] are some of the important methods available to compute exact solutions of nonlinear PDEs. Although there is no universal strategy for solving nonlinear PDEs, Lie symmetry analysis is one of the most effective and reliable techniques for discovering new exact solutions to nonlinear PDEs arising in applied mathematics and physics. For the application of some well-known methods to compute numerical and exact solutions of differential equations, the interested reader is referred to see [20–25].

In this paper, the Lie point symmetry method is applied to nonlinear systems. The symmetry reductions related to the nonlocal symmetry can be analyzed in [26] where the truncated Painlevé analysis or the Möbious invariant form yields the nonlocal symmetry of the Gardner equation. Here, the extended system locates the nonlocal symmetry to the local Lie point symmetries. So the nonlocal symmetry is used to find possible reductions in symmetry using the

**Data Availability Statement:** The article does not report data and the data availability policy is not applicable to my article.

**Competing interests:** The authors have declared that no competing interests exist.

localization technique. In [27] the nonlocal symmetries for the (2+1)-dimensional Konopel-chenko–Dubrovsky equation are determined with the shortened Painlevé method and the Möbious (conformal) invariant form. Here the nonlocal symmetries are reduced to the Lie point symmetries by inserting auxiliary dependent variables. So, we can generate finite symmetry transformations by solving the initial value problem of the prolonged systems. Similarly using the truncated Painlevé approach and the Möbius (conformal) invariant form, we can drive the nonlocal symmetry for the Drinfel'd-Sokolov-Wilson equation in [28]. Meanwhile, for the use of symmetry reductions related to nonlocal symmetry, the nonlocal symmetry will be localized to the Lie point symmetry by introducing three dependent variables.

Recently, Li, Tian, Yang, and Fan have done some interesting work in deriving the solutions of the Wadati-Konno-Ichikawa equation and complex short pulse equation with the help of the Dbar-steepest descent method. They solved the long-time asymptotic behavior of the solutions of these equations and proved the soliton resolution conjecture and the asymptotic stability of solutions of these equations. (See: [29–31]).

The $(1 + 1)$- dimensional integro-differential Ito equation is a well known NLPDE, It can be governed by

$$u_{tt} + u_{xxxt} + 3\left(2 u_x u_t + u u_{xt}\right) + 3 u_{xx} \partial_x^{-1}(u_t) = 0.$$

The mathematical representation of first integro-differential KP hierarchy equation is

$$u_t - \frac{1}{2} u_{xxy} - \frac{1}{2} \partial_x^{-2}\left(u_{yyy}\right) - 2 u_x \partial_x^{-1}\left(u_y\right) - 4 u u_y = 0.$$

The Calogero—Bogoyavlenskii—schiff (CBS) equation is represented by

$$v_t + v_{xxy} + 4 v v_y + 2 v_x \partial_x^{-1}\left(v_y\right) = 0.$$

The modified Calogero-Bogoyavlenskii-schiff (CBS) equation can be expressed symbolically as

$$v_t + v_{xxy} + 4 v^2 v_y + 4 v_x \partial_x^{-1}\left(v v_y\right) = 0.$$

The standard form of modified KdV-CBS equation is

$$u_t - 4 u^2 u_y - 2 u_x \partial^{-1}\left(u^2\right)_y + u_{xxy} - 6 u^2 u_x + u_{xxx} = 0.$$

Developing techniques for the exact solutions of these models, which entail systems of nonlinear PDEs, has been important in the study of nonlinear PDEs.

In this article, we compute the exact solutions of five well-known NLPDEs, namely, the $(1 + 1)$-dimensional integro-differential Ito equation, the first integro-differential KP hierarchy equation, the Calogero-Bogoyavlenskii-Schiff (CBS) equation, the modified Calogero-Bogoyavlenskii-Schiff (CBS) equation, and the modified KdV-CBS equation. The classical Lie point symmetries are utilized to reduce the number of independent variables via similarity transformations, which ultimately give rise to the exact solutions of the prescribed equations. The sine-cosine method is also employed to derive general exact solutions. The exact solutions presented in this study are concise and straightforward and can be used to establish new solutions for other kinds of NLPDEs arising in different areas of mathematical physics.

The paper is organized in the following pattern: In Section 2, basic definitions, important relations, and the fundamental theorem of the sine-cosine method are presented. In Section 3, Lie symmetries and exact solutions of $(1 + 1)$-dimensional integro-differential Ito equations are constructed. In Section 4, Lie symmetries and exact solutions of the first integro-

differential KP hierarchy equation are determined. In Section 5, Lie symmetries and exact solutions of the Calogero-Bogoyavlenskii-Schiff (CBS) equations are analyzed. In section 6, Lie symmetries and exact solutions of the modified Calogero-Bogoyavlenskii-Schiff (CBS) and in section 7, Lie symmetries and exact solutions of the modified Kdv-CBS equations are evaluated. In the last section, we summarize the concluding remarks.

## 2 Fundamental operators

Consider the following $p^{th}$ order system of differential equations

$$E_{\gamma}(x, u, u^{(1)}, u^{(2)}, ..., u^{(p)}) = 0, \quad \gamma = 1, 2, 3, ..., l, \tag{2.1}$$

with $m$ independent variables $x = (x^1, x^2, x^3, \ldots, x^m)$ and $n$ dependent variables $u = (u^1, u^2, u^3 \ldots, u^n)$.

The differential function $E_{\gamma}(x, u, u^{(1)}, \ldots, u^{(p)})$, in (2.1), is a $p^{th}$ order differential invariant of a group G if

$$X^{[p]}(E) = 0, \tag{2.2}$$

where $X^{[p]}$ is the $p^{th}$ prolongation of the Lie-Bäclund or generalized operator $X$ defined by

$$X = \left(\frac{\partial}{\partial x^i}\right) \xi^i(x, u) + \left(\frac{\partial}{\partial u^{\beta}}\right) \eta^{\beta}(x, u) + \sum_{q \geq 1} \theta^{\beta}_{i_1, i_2, i_3, ..., i_q} \frac{\partial}{\partial u^{\beta}_{i_1 ... i_q}}, \tag{2.3}$$

where $\theta_{i_1, i_2, i_3, ..., i_q}$ can be determined from

$$\theta^{\beta}_i = D_i(\eta^{\beta}) - u^{\beta}_j D_i(\xi^j),$$

$$\theta^{\beta}_{i_1, i_2} = D_{i_2}(\theta^{\beta}_{i_1}) - u^{\beta}_{j i_1} D_{i_2}(\xi^j),$$

$$\vdots$$

$$\theta^{\beta}_{i_1, i_2, i_3, ..., i_q} = D_{i_q}(\theta^{\beta}_{i_1, i_2, i_3, ..., i_{q-1}}) - u^{\beta}_{j i_1, i_2, i_3, ... i_{q-1}} D_{i_q}(\xi^j), \quad q > 1.$$

The total derivative operator with respect to $x_i$ takes the form,

$$D_i = \frac{\partial}{\partial x^i} + u^{\beta}_i \frac{\partial}{\partial u^{\beta}} + u^{\beta}_{ij} \frac{\partial}{\partial u^{\beta}_j} + \cdots. \tag{2.4}$$

The invariants of the differential function $E_{\gamma}$ in (2.1) can be obtained by solving characteristic equations derived from Eq (2.2).

The Euler operator is defined as

$$\frac{\delta}{\delta u^{\beta}} = \frac{\partial}{\partial u^{\beta}} - D_i \frac{\partial}{\partial u^{\beta}_i} + D_i D_j \frac{\partial}{\partial u^{\beta}_{ij}} - \cdots, \tag{2.5}$$

where $D_i$ is the total derivative operator.

### 2.1 Sine-Cosine method

There is no particular method that works for all types of nonlinear evolution equations. The sine-cosine method ([32–34]) is one such technique and it can be used to solve a wide variety of nonlinear evolution equations. The sine-cosine method works for those PDEs that admit translational symmetries. If a PDE possesses translational symmetries, we may convert it to an

ODE by introducing a wave variable and assuming that the solution would take the form of a sine function or cosine function. The sine-cosine algorithm is described as follows:

**Sine-Cosine algorithm:**

- Integrate the ODE $Q(u, u_s, u_{ss}, u_{sss}, \ldots) = 0$ as many times as possible and set the constants of integration to zero.

- Suppose the solution of the form

$$u(s) = \alpha \sin(\beta s)^k,\qquad (2.6)$$

or

$$u(s) = \alpha \cos(\beta s)^k,\qquad (2.7)$$

where we need to find the parameters $\alpha$, $\beta$ and $k$.

- Substitute (2.6) or (2.7) in $Q(u, u_s, u_{ss}, u_{sss}, \ldots) = 0$ and balance the terms of sine functions when (2.6) is used or balance the terms of cosine functions when (2.7) is utilized.

- After defining the value of $k$, we separate the terms concerning powers of cosine or sine functions to obtain algebraic system of equations in terms of $\alpha$ and $\beta$.

After computing the values of $\alpha$ and $\beta$ and inserting them into the main equation, we get at the solution.

## 3 Lie symmetries and exact solutions of (1 + 1)- dimensional Integro-differential Ito equation

The (1 + 1)- dimensional integro-differential Ito equation is governed by

$$u_{tt} + u_{xxxt} + 3\,(\,2\,u_x\,u_t + u\,u_{xt}) + 3\,u_{xx}\,\partial_x^{-1}(u_t) = 0.\qquad (3.1)$$

Eq (3.1) can be expressed as

$$u_{tt} + u_{xxxt} + 3\,(\,2\,u_x\,u_t + u\,u_{xt}) + 3\,u_{xx}\,v = 0,$$

$$\qquad (3.2)$$

$$u_t = v_x,$$

where $v = \partial_x^{-1}(\,u_t)\ \ and\ \ \partial^{-1}$ denotes the integral with respect to the subscripts.

Using Eq (2.2), the following overdetermined linear system of PDEs is obtained

$$\xi_t^1 = 3\ \xi_x^2, \qquad \xi_u^1 = 0, \qquad \xi_v^1 = 0, \qquad \xi_x^1 = 0, \qquad \xi_t^2 = 0, \qquad \xi_u^2 = 0,$$

$$\xi_v^2 = 0, \qquad \xi_{xx}^2 = 0, \qquad \eta^1 = -2\ \xi_x^2\ u, \qquad \eta^2 = -4\ \xi_x^2\ v.$$

The following Lie point symmetries can be generated by solving determining equations with one component equal to one and the remaining equal to zero

$$X_1 = \frac{\partial}{\partial x}\,, \qquad X_2 = \frac{\partial}{\partial t}, \qquad X_3 = 3\,t\,\frac{\partial}{\partial t} + x\,\frac{\partial}{\partial x} - 2\,u\,\frac{\partial}{\partial u} - 4\,v\,\frac{\partial}{\partial v}.$$

The translational symmetries corresponding to system (3.2) are

$$X_1 = \frac{\partial}{\partial x}\,, \qquad X_2 = \frac{\partial}{\partial t}.\qquad (3.3)$$

The following similarity transformations are obtained using combination of $X_1$ and $X_2$ i.e $X = X_1 + \alpha X_2$

$$r = -\alpha\, t + x, \quad s = t, \quad u(t,x) = v(r), \quad u_t = -\alpha\, u_r,$$

which further implies

$$u_{tt} = \alpha^2\, u_{rr}, \quad u_{tx} = -\alpha\, u_{rr}, \quad u_x = u_r, \quad u_{xx} = u_{rr},$$
$$u_{xxx} = u_{rrr}, \quad v_t = -\alpha\, v_r, \quad u_{xxxt} = -\alpha\, u_{rrrr}. \tag{3.4}$$

Substituting (3.4) in system (3.2), we obtain

$$\alpha^2\, u_{rr} - \alpha\, u_{rrrr} + 6\, u_r\, (-\alpha\, u_r) + 3\, u\, (-\alpha\, u_{rr}) + 3\, u_{rr}\, v = 0,$$
$$\tag{3.5}$$
$$v = c_1 - \alpha\, u.$$

The system (3.5) gives rise to

$$\alpha^2\, u_{rr} - \alpha\, u_{rrrr} - 6\, \alpha\, u_r^2 + 3\, c_1\, u_{rr} - 6\, \alpha\, u\, u_{rr} = 0.$$

Integrating twice yields

$$(\alpha^2 + 3\, c_1)\, u - 3\, \alpha\, u^2 - \alpha\, u_{rr} = 0. \tag{3.6}$$

The particular solution of (3.6) is

$$u(r) = -\frac{1}{2}\left[ \alpha\, tanh\left(\frac{1}{2\,\alpha}\delta\,(c_1 - r)\right)^2 + \frac{3}{\alpha}\, tanh\left(\frac{1}{2\,\alpha}\delta\,(c_1 - r)\right)^2 c_1 - \frac{\alpha^2 + 3\, c_1}{\alpha}\right],$$
$$\tag{3.7}$$
$$v(r) = c_1 + \frac{\alpha}{2}\left[ \alpha\, tanh\left(\frac{1}{2\,\alpha}\delta\,(c_1 - r)\right)^2 + \frac{3}{\alpha}\, tanh\left(\frac{1}{2\,\alpha}\delta\,(c_1 - r)\right)^2 c_1 - \frac{\alpha^2 + 3\, c_1}{\alpha}\right].$$

The set of solutions (3.7) expressed in terms of original variables are

$$u(t,x) = -\frac{1}{2}\left[ \alpha\, tanh\left(\frac{1}{2\,\alpha}\delta\,(c_1 - (x - \alpha\, t))\right)^2 + \frac{3}{\alpha}\, tanh\left(\frac{1}{2\,\alpha}\delta\,(c_1 - (x - \alpha\, t))\right)^2 c_1 - \frac{\alpha^2 + 3\, c_1}{\alpha}\right],$$

and

$$v(t,x) = c_1 + \frac{\alpha}{2}\left[ \alpha\, tanh\left(\frac{1}{2\,\alpha}\delta\,(c_1 - (x - \alpha\, t))\right)^2 + \frac{3}{\alpha}\, tanh\left(\frac{1}{2\,\alpha}\delta\,(c_1 - (x - \alpha\, t))\right)^2 c_1 - \frac{\alpha^2 + 3\, c_1}{\alpha}\right],$$

where $\delta = \sqrt{\alpha^3 + 3\, \alpha\, c_1}$.

Now using the sine-cosine approach [32–34], we obtain exact solutions of Eq (3.6). Suppose (3.6) has a solution such as

$$u(r) = \lambda\, cos^k(\omega\, r). \tag{3.8}$$

Substituting the values of $u$ from (3.8) in (3.6) yields

$$\alpha^2\, \lambda\, cos^k(\omega\, r) - \alpha\, \frac{\partial^2}{\partial r^2}\left(\lambda\, cos^k(\omega\, r)\right) - 3\, \alpha\, \lambda^2(cos^k(\omega\, r))^2 + 3\, \lambda\, cos^k(\omega\, r)\, c_1 = 0. \tag{3.9}$$

Eq (3.9) is satisfied if

$$2k = k - 2.$$

Substituting $k = -2$ in (3.9), we obtain

$$\frac{4\lambda\alpha\omega^2}{cos^2(\omega r)} - \frac{3\lambda^2\alpha}{cos^4(\omega r)} - \frac{6\lambda\alpha\omega^2}{cos^4(\omega r)} + \frac{\lambda\alpha^2}{cos^2(\omega r)} + \frac{3\lambda c_1}{cos^2(\omega r)} = 0.$$

Comparing the coefficients of powers of $\dfrac{1}{cos(\omega r)}$, we have

$$4\lambda\alpha\omega^2 + \alpha^2\lambda + 3\lambda c_1 = 0,$$

$$-3\alpha\lambda^2 - 6\alpha\lambda\omega^2 = 0. \tag{3.10}$$

Solution of (3.10) gives

$$\lambda = \frac{\alpha^2 + 3 c_1}{2\alpha}, \quad \omega = \sqrt{-\frac{\alpha^2 + 3 c_1}{4\alpha}}.$$

Eq (3.8) using the value of $\lambda$, $\omega$ and $k$ results in

$$u(r) = \frac{\alpha^2 + 3 c_1}{2\alpha} sec^2\left(\sqrt{-\frac{\alpha^2 + 3 c_1}{4\alpha}}\, r\right),$$

where $r = x - \alpha t$. The solution of (3.6) in terms of original variables can be expressed as

$$u(t,x) = \frac{\alpha^2 + 3 c_1}{2\alpha} sec^2\left(\sqrt{-\frac{\alpha^2 + 3 c_1}{4\alpha}}\,(x - \alpha t)\right),$$

$$v(t,x) = c_1 - \frac{\alpha^2 + 3 c_1}{2} sec^2\left(\sqrt{-\frac{\alpha^2 + 3 c_1}{4\alpha}}\,(x - \alpha t)\right),$$

which constitute the exact solutions of (1 + 1)- dimensional integro-differential Ito equation (3.2).

## 4 Lie symmetries and exact solutions of the first integro-differential KP hierarchy equation

The first integro-differential KP hierarchy equation is governed by

$$u_t - \frac{1}{2} u_{xxy} - \frac{1}{2}\partial_x^{-2}(u_{yyy}) - 2 u_x \partial_x^{-1}(u_y) - 4 u u_y = 0, \tag{4.1}$$

which can be rewritten as

$$u_t - \frac{1}{2} u_{xxy} - \frac{1}{2} v_{yy} - 2 u_x v_x - 4 u u_y = 0,$$

$$u_y = v_{xx}, \tag{4.2}$$

where $v_x = \partial_x^{-1}(u_y)$ and $\partial^{-1}$ denotes the integral with respect to the subscripts.

Using Eq (4.2), the following overdetermined linear set of PDEs are obtained

$$\eta^2_u = 0, \qquad \eta^2_v = -\frac{1}{2}\ \xi^1_t, \qquad \eta^2_x = -\frac{1}{2}\ \xi^2_t, \qquad \eta^2_{yy} = -\frac{1}{2}\ \xi^3_{tt}, \qquad \xi^1_u = 0, \quad \xi^1_v = 0,$$

$$\xi^1_x = 0, \qquad \xi^1_y = 0, \qquad \xi^2_u = 0, \qquad \xi^2_v = 0, \qquad \xi^2_x = \frac{1}{4}\ \xi^1_t, \qquad \xi^2_y = 0,$$

$$\xi^3_u = 0, \qquad \xi^3_v = 0, \qquad \xi^3_x = 0, \qquad \xi^3_y = \frac{1}{2}\ \xi^1_t, \qquad \eta^1 = -\frac{1}{2}\ u\ \xi^1_t - \frac{1}{4}\ \xi^3_t.$$

The following Lie point symmetries can be generated by solving determining equations with one component equal to one and the remaining equal to zero

$$X_1 = \frac{\partial}{\partial t}, \qquad X_2 = \frac{\partial}{\partial x}, \qquad X_3 = \frac{\partial}{\partial y}, \qquad X_4 = y\frac{\partial}{\partial v},$$

$$X_5 = \frac{\partial}{\partial v}, \qquad X_6 = t\frac{\partial}{\partial v}, \qquad X_7 = t\frac{\partial}{\partial y} - \frac{1}{4}\frac{\partial}{\partial u},$$

$$X_8 = ty\frac{\partial}{\partial v}, \qquad X_9 = t\frac{\partial}{\partial x} - \frac{1}{2}x\frac{\partial}{\partial v}, \qquad X_{10} = t\frac{\partial}{\partial t} + \frac{1}{4}x\frac{\partial}{\partial x} + \frac{1}{2}y\frac{\partial}{\partial y} - \frac{1}{2}u\frac{\partial}{\partial u} - \frac{1}{2}v\frac{\partial}{\partial v}.$$

Using the combination of translational symmetries

$$X = X_1 + X_2 + \alpha\ X_3$$

or

$$X = \frac{\partial}{\partial t} + \frac{\partial}{\partial x} + \alpha\ \frac{\partial}{\partial y}.$$

We obtain the similarity transformations

$$r = y - \alpha\ t, \quad s = x - t, \quad q = t, \quad u_y = u_r, \quad u_t = -\alpha\ u_r - u_s, \quad u_x = u_s.$$

Substituting above values in Eq (4.2), we obtain

$$-\alpha\ u_r - u_s - \frac{1}{2}\ u_{ssr} - \frac{1}{2}\ v_{rr} - 2\ u_s\ v_s - 4\ u\ u_r = 0. \tag{4.3}$$

$$u_r = v_{ss}.$$

The system (4.3) admits the Lie point symmetries $X_1 = \frac{\partial}{\partial r}$, $X_2 = \frac{\partial}{\partial s}$. Using the combination of symmetries $X_1$ and $X_2$, we have

$$X = \frac{\partial}{\partial r} + \beta\ \frac{\partial}{\partial s}.$$

The similarity transformations are

$$g = s - \beta\ r, \quad h = r, \quad u_r = -\beta\ u_g, \quad u_s = u_g, \quad u_{rr} = \beta^2\ u_{gg}, \quad u_{ss} = u_{gg}. \tag{4.4}$$

Substituting (4.4) in system (4.3), we obtain

$$\alpha \beta\, u_g - u_g + \frac{\beta}{2}\, u_{ggg} - \frac{\beta^2}{2} v_{gg} - 2\, u_g\, v_g + 4\, \beta\, u\, u_g = 0,$$

$$v_g = c_1 - \beta\, u.$$

(4.5)

The system (4.5) gives rise to

$$\alpha \beta\, u_g - u_g + \frac{\beta}{2}\, u_{ggg} + \frac{\beta^3}{2} u_g - 2\, c_1\, u_g + 6\, \beta\, u\, u_g = 0.$$

Integration w.r.t g, yields

$$\frac{\beta}{2}\, u_{gg} + 3\, \beta\, u^2 + (\alpha\,\beta + \frac{\beta^3}{2} - 2\, c_1 - 1)u = 0.$$

(4.6)

The particular solution of (4.6) is

$$u(g) = -\frac{1}{4\,\beta}\left[ \beta^3 \tan\left(\frac{\delta(\,c_1 - g)}{2\,\beta}\right)^2 + 2\,\alpha\,\beta\, \tan\left(\frac{\delta(\,c_1 - g)}{2\,\beta}\right)^2 + \beta^3 \right.$$

$$\left. -4\,\tan\left(\frac{\delta\,(\,c_1 - g)}{2\,\beta}\right)^2 c_1 + 2\,\alpha\,\beta - 2\,\tan\left(\frac{\delta\,(\,c_1 - g)}{2\,\beta}\right)^2 - 4\,c_1 - 2 \right],$$

$$v(g) = -\frac{1}{2}\,\frac{\beta^4}{\delta}\, \tan\left(\frac{\delta\,(\,c_1 - g)}{2\,\beta}\right) + \frac{1}{2}\,\frac{\beta^4}{\delta}\, \arctan\left[\tan\left(\frac{\delta\,(\,c_1 - g)}{2\,\beta}\right)\right]$$

$$-\frac{\alpha\,\beta^2}{\delta}\, \tan\left(\frac{\delta\,(\,c_1 - g)}{2\,\beta}\right) + \frac{\alpha\,\beta^2}{\delta}\, \arctan\left[\tan\left(\frac{\delta\,(\,c_1 - g)}{2\,\beta}\right)\right]$$

$$+\frac{1}{4}\,\beta^3\, g + \frac{2c_1\,\beta}{\delta}\, \tan\left(\frac{\delta\,(\,c_1 - g)}{2\,\beta}\right) - \frac{2\,c_1\,\beta}{\delta}\, \arctan\left[\tan\left(\frac{\delta\,(\,c_1 - g)}{2\,\beta}\right)\right]$$

$$+\frac{1}{2}\,\alpha\,\beta\, g + \frac{\beta}{\delta}\,\tan\left(\frac{\delta\,(\,c_1 - g)}{2\,\beta}\right) - \frac{\beta}{\delta}\, \arctan\left[\tan\left(\frac{\delta\,(\,c_1 - g)}{2\,\beta}\right)\right] - \frac{1}{2}.$$

The above solutions can be finally expressed in terms of original variables as

$$u(t,x,y) = -\frac{1}{4\,\beta}\left[ \beta^3 \tan\left(\frac{\delta\,(\,c_1 + \beta\,y - x - t\,(\,\alpha\,\beta - 1))}{2\,\beta}\right)^2 + 2\,\alpha\,\beta\, \tan\left(\frac{\delta\,(\,c_1 + \beta\,y - x - t\,(\,\alpha\,\beta - 1))}{2\,\beta}\right)^2 + \beta^3 \right.$$

$$\left. -4\,\tan\left(\frac{\delta\,(\,c_1 + \beta\,y - x - t\,(\,\alpha\,\beta - 1))}{2\,\beta}\right)^2 c_1 + 2\,\alpha\,\beta - 2\,\tan\left(\frac{\delta\,(\,c_1 + \beta\,y - x - t\,(\,\alpha\,\beta - 1))}{2\,\beta}\right)^2 - 4\,c_1 - 2 \right],$$

$$v(t,x,y) = -\frac{1}{2}\,\frac{\beta^4}{\delta}\tan\left(\frac{\delta\,(\,c_1 + \beta\,y - x - t\,(\,\alpha\,\beta - 1))}{2\,\beta}\right)^2 + \frac{1}{2}\,\frac{\beta^4}{\delta}\, \arctan\left[\tan\left(\frac{\delta\,(\,c_1 + \beta\,y - x - t\,(\,\alpha\,\beta - 1))}{2\,\beta}\right)\right]$$

$$-\frac{\alpha\,\beta^2}{\delta}\, \tan\left(\frac{\delta\,(\,c_1 + \beta\,y - x - t\,(\,\alpha\,\beta - 1))}{2\,\beta}\right) + \frac{\alpha\,\beta^2}{\delta}\, \arctan\left[\tan\left(\frac{\delta\,(\,c_1 + \beta\,y - x - t\,(\,\alpha\,\beta - 1))}{2\,\beta}\right)\right]$$

$$+\frac{2\,c_1\,\beta}{\delta}\, \tan\left(\frac{\delta\,(\,c_1 + \beta\,y - x - t\,(\,\alpha\,\beta - 1))}{2\,\beta}\right) - \frac{2\,c_1\,\beta}{\delta}\, \arctan\left[\tan\left(\frac{\delta\,(\,c_1 + \beta\,y - x - t\,(\,\alpha\,\beta - 1))}{2\,\beta}\right)\right]$$

$$+\frac{\beta}{\delta}\, \tan\left(\frac{\delta\,(\,c_1 + \beta\,y - x - t\,(\,\alpha\,\beta - 1))}{2\,\beta}\right) - \frac{\beta}{\delta}\, \arctan\left[\tan\left(\frac{\delta\,(\,c_1 + \beta\,y - x - t\,(\,\alpha\,\beta - 1))}{2\,\beta}\right)\right]$$

$$+\frac{1}{4}\,\beta^3\,(-\beta\,y + x + t\,(\alpha\,\beta - 1)) + \frac{1}{2}\,\alpha\,\beta\,(\,-\beta\,y + x + t\,(\,\alpha\,\beta - 1)) + \frac{1}{2}\,\beta\,y - \frac{1}{2}\,x - \frac{1}{2}\,t\,(\,\alpha\,\beta - 1),$$

where $\delta = \sqrt{\beta^4 + 2\alpha\beta^2 - 4\beta c_1 - 2\beta}$.

Now, Using the sine-cosine approach [32–34], we obtain the explicit solutions of Eq (4.6). Suppose (4.6) has a solution such as

$$u(g) = \lambda\, cos^k(\omega\, g). \tag{4.7}$$

Substituting the value of $u$ from (4.7) in (4.6) yields

$$\left(\alpha\,\beta - 1 + \frac{1}{2}\,\beta^3 - 2\,c_1\right)\lambda\,cos^k(\omega\,g) + \frac{1}{2}\,\beta\left(\frac{\partial^2}{\partial g^2}\,(\lambda\,cos^k(\omega\,g))\right) + 3\,\beta\,\lambda^2(cos^k(\omega\,g))^2 = 0. \tag{4.8}$$

Eq (4.8) is satisfied if $2k = k - 2$. Replacing $k = -2$ in (4.8) to obtain

$$-\frac{2\,\lambda\,\beta\,\omega^2}{cos^2(\omega\,g)} + \frac{3\,\beta\,\lambda^2}{cos^4(\omega\,g)} + \frac{1}{2}\frac{\lambda\,\beta^3}{cos^2(\omega\,g)} + \frac{3\,\beta\,\lambda\,\omega^2}{cos^4(\omega\,g)} + \frac{\lambda\,\alpha\,\beta}{cos^2(\omega\,g)} - \frac{2\,\lambda\,c_1}{cos^2(\omega\,g)} - \frac{\lambda}{cos^2(\omega\,g)} = 0.$$

Comparing the coefficients of powers of $\dfrac{1}{cos(\omega\,g)}$, we get

$$-2\,\beta\,\omega^2 + \frac{1}{2}\,\beta^3 + \alpha\,\beta - 2\,c_1 - 1 = 0,$$
$$\tag{4.9}$$
$$3\,\beta\,\lambda + 3\,\beta\,\omega^2 = 0.$$

Solving (4.9) gives

$$\lambda = \frac{1 - \alpha\,\beta - \frac{1}{2}\,\beta^3 + 2\,c_1}{2\,\beta}, \quad \omega = \sqrt{\frac{\alpha\,\beta - 1 + \frac{1}{2}\,\beta^3 - 2\,c_1}{2\,\beta}}.$$

Using the values of $\lambda$, $\omega$ and $k$ in (4.7) result in

$$u(g) = \frac{1 - \alpha\,\beta - \frac{1}{2}\,\beta^3 + 2\,c_1}{2\,\beta}\;sec^2\left(\sqrt{\frac{\alpha\,\beta - 1 + \frac{1}{2}\,\beta^3 - 2\,c_1}{2\,\beta}}\;g\right).$$

The solution of (4.6) in terms of original variables can be expressed as

$$u(t,x,y) = \frac{1 - \alpha\,\beta - \frac{1}{2}\,\beta^3 + 2\,c_1}{2\,\beta}\;sec^2\left(\sqrt{\frac{\alpha\,\beta - 1 + \frac{1}{2}\,\beta^3 - 2\,c_1}{2\,\beta}}\;(x - \beta\,y + t\,(\alpha\,\beta - 1))\right)$$

and this constitute the exact solutions of the first integro-differential KP hierarchy equation (4.1).

## 5 Lie symmetries and exact solutions of the Calogero-Bogoyavlenskii-schiff (CBS) equation

The Calogero—Bogoyavlenskii—schiff (CBS) equation is governed by

$$v_t + v_{xxy} + 4\,v\,v_y + 2\,v_x\,\partial_x^{-1}(v_y) = 0. \tag{5.1}$$

Eq (5.1) can be expressed as system of following two equations

$$v_t + v_{xxy} + 4\,vv_y + 2\,v_x\,u = 0,$$

$$u_x = v_y, \tag{5.2}$$

where $u = \partial_x^{-1}(\,v_y)$ $\quad and \quad \partial^{-1}$ denotes the integral with respect to the subscripts.

Using Eq (5.2), the following overdetermined linear set of PDEs are obtained

$$\xi_u^1 = 0, \qquad \xi_v^1 = 0, \qquad \xi_x^1 = 0, \qquad \xi_y^1 = 0, \qquad \xi_{ttt}^1 = 0, \qquad \xi_u^2 = 0,$$

$$\xi_v^2 = 0, \qquad \xi_y^2 = 0, \qquad \xi_{tx}^2 = \frac{1}{4}\,\xi_{tt}^1, \qquad \xi_{xx}^2 = 0, \qquad \xi_u^3 = 0, \qquad \xi_v^3 = 0,$$

$$\xi_x^3 = 0, \qquad \xi_y^3 = -2\quad \xi_x^2 + \xi_t^1, \qquad \xi_{tt}^3 = 0, \qquad \eta^1 = \xi_x^2\,u - u\,\xi_t^1 + \frac{1}{2}\,\xi_t^2, \qquad \eta^2 = -2\,\xi_x^2\,v + \frac{1}{4}\,\xi_t^3.$$

The following Lie point symmetries can be generated by solving determining equations with one component equal to one and the remaining equal to zero

$$X_1 = \frac{\partial}{\partial t} + \frac{\partial}{\partial x}, \qquad X_2 = \frac{\partial}{\partial x} + \frac{\partial}{\partial y}, \qquad X_3 = \frac{\partial}{\partial x} + t\,\frac{\partial}{\partial y} + \frac{1}{4}\,\frac{\partial}{\partial v},$$

$$X_4 = t\,\frac{\partial}{\partial t} + \frac{\partial}{\partial x} + y\,\frac{\partial}{\partial y} - u\,\frac{\partial}{\partial u}, \qquad X_5 = (1+x)\,\frac{\partial}{\partial x} - 2\,y\,\frac{\partial}{\partial y} + u\,\frac{\partial}{\partial u} - 2\,v\,\frac{\partial}{\partial v},$$

$$X_6 = \frac{1}{2}\,t^2\,\frac{\partial}{\partial t} + \left(1 + \frac{1}{4}\,t\,x\right)\,\frac{\partial}{\partial x} + \frac{1}{2}\,y\,t\,\frac{\partial}{\partial y} + \left(-\frac{3}{4}\,t\,u + \frac{1}{8}\,x\right)\,\frac{\partial}{\partial u} + \left(-\frac{1}{2}\,v\,t + \frac{1}{8}\,y\right)\,\frac{\partial}{\partial v}.$$

The combination of translational symmetries corresponding to system (5.2) are

$$X = X_1 + X_2 + \alpha\,X_3.$$

or

$$X = \frac{\partial}{\partial t} + \frac{\partial}{\partial x} + \alpha\,\frac{\partial}{\partial y}.$$

The similarity transformations are obtained using combination of $X = X_1 + X_2 + \alpha X_3$

$$r = y - \alpha\,t, \quad s = x - t, \quad q = t, \quad u_y = u_r, \quad u_t = -\alpha\,u_r - u_s, \quad u_x = u_s. \tag{5.3}$$

Substituting values from (5.3) in Eq (5.2), we obtain

$$-\alpha\,v_r - v_s + v_{ssr} + 4\,v\,v_r + 2\,v_s\,u = 0,$$

$$u_s = v_r. \tag{5.4}$$

The system (5.4) admits the translational symmetries $X_1 = \dfrac{\partial}{\partial r}$, $X_2 = \dfrac{\partial}{\partial s}$. We use the combination of $X_1$ and $X_2$, i.e

$$X = \frac{\partial}{\partial r} + \beta \frac{\partial}{\partial s}$$

and compute the similarity transformations

$$g = s - \beta\, r, \quad h = r, \quad u_r = -\beta\, u_g, \quad u_s = u_g, \quad u_{rr} = \beta^2\, u_{gg}, \quad u_{ss} = u_{gg}. \tag{5.5}$$

Substituting (5.5) in Eq (5.4) gives rise to

$$(\alpha\,\beta - 1 + 2\,u\,)\,v_g - \beta\,v_{ggg} - 4\,\beta\,v\,v_g = 0,$$

$$\tag{5.6}$$

$$-\beta\,v_g - u_g = 0,$$

where

$$u = c_1 - \beta\,v.$$

From system (5.6), we conclude

$$-\beta\,v_{ggg} - 6\,\beta\,v\,v_g + (\alpha\,\beta - 1 + 2\,c_1\,)\,v_g = 0.$$

Integrating w.r.t g and choosing the constant of integration to zero, we arrive at

$$-\beta\,v_{gg} - 3\,\beta v^2 + (\,\alpha\,\beta - 1 + 2\,c_1)\,v = 0, \tag{5.7}$$

which finally yields

$$v(g) = -\frac{1}{2\,\beta}\left[\alpha\,\beta\,tanh\left(\frac{\delta\,(\,c_1 - g)}{2\,\beta}\right)^2 + 2\,tanh\left(\frac{\delta\,(\,c_1 - g)}{2\,\beta}\right)^2 c_1 - \alpha\,\beta - tanh\left(\frac{\delta\,(c_1 - g)}{2\,\beta}\right)^2 - 2\,c_1 + 1\right].$$

Using $v(g)$ in system (5.6), we obtain

$$u(g) = c_1 + \frac{1}{2}\left[\alpha\,\beta\,tanh\left(\frac{\delta\,(\,c_1 - g)}{2\,\beta}\right)^2 + 2\,tanh\left(\frac{\delta\,(\,c_1 - g)}{2\,\beta}\right)^2 c_1 - \alpha\,\beta - tanh\left(\frac{\delta\,(c_1 - g)}{2\,\beta}\right)^2 - 2\,c_1 + 1\right].$$

We apply the inverse informations (5.3), the solution can be expressed in original variables as

$$v(t, x, y) = -\frac{1}{2\,\beta}\left[\alpha\,\beta\,tanh\left(\frac{\delta\,(\,c_1 - x + \beta\,y - t\,(\,\alpha\,\beta - 1))}{2\,\beta}\right)^2 - \alpha\,\beta - 2\,c_1 + 1\right.$$

$$\left. + 2\,tanh\left(\frac{\delta\,(\,c_1 - x + \beta\,y - t\,(\,\alpha\,\beta - 1))}{2\,\beta}\right)^2 c_1 - tanh\left(\frac{\delta\,(c_1 - x + \beta\,y - t\,(\,\alpha\,\beta - 1))}{2\,\beta}\right)^2\right],$$

$$u(t, x, y) = \frac{1}{2}\,\alpha\,\beta\,tanh\left(\frac{\delta\,(\,c_1 - x + \beta\,y - t\,(\,\alpha\,\beta - 1))}{2\,\beta}\right)^2 - \frac{1}{2}\,\alpha\,\beta + \frac{1}{2}$$

$$+ tanh\left(\frac{\delta(c_1 - x + \beta\,y - t\,(\alpha\,\beta - 1))}{2\,\beta}\right)^2 c_1 - \frac{1}{2}\,tanh\left(\frac{\delta\,(\,c_1 - x + \beta\,y - t\,(\,\alpha\,\beta - 1))}{2\,\beta}\right)^2,$$

where $\delta = \sqrt{\alpha\,\beta^2 + 2\,\beta\,c_1 - \beta}$.

which constitute an exact solution of Eq (5.1).

Now, Using the sine-cosine approach [32–34], we obtain the explicit solutions of Eq (5.7). Suppose (5.7) has a solution such as

$$v(g) = \lambda \ cos^k(\omega g). \tag{5.8}$$

Substituting $v$ from (5.8) in (5.7), yields

$$(\alpha \ \beta + 2 \ c_1 - 1) \ \lambda \ cos^k \ (\omega \ g) - \beta \ \frac{\partial^2}{\partial g^2} \ (\lambda \ cos^k \ (\omega \ g)) - 3 \ \beta \ \lambda^2 \ (cos^k \ (\omega \ g))^2 = 0. \tag{5.9}$$

Eq (5.9) is satisfied if $2k = k - 2$. Substituting $k = -2$ in (5.9) to obtain

$$\frac{4 \ \lambda \ \beta \ \omega^2}{cos^2 \ (\omega \ g)} - \frac{3 \ \beta \ \lambda^2}{cos^4 \ (\omega \ g)} - \frac{6 \ \beta \ \lambda \ \omega^2}{cos^4 \ (\omega \ g)} + \frac{\lambda \ \alpha \ \beta}{cos^2 \ (\omega \ g)} + \frac{2 \ \lambda \ c_1}{cos^2 \ (\omega \ g)} - \frac{\lambda}{cos^2 \ (\omega \ g)} = 0.$$

Comparing the coefficients of powers of $\frac{1}{cos(\omega \ g)}$, we obtain the following system

$$4 \ \beta \ \omega^2 + \alpha \ \beta + 2 \ c_1 - 1 = 0,$$

$$\lambda + 2 \ \omega^2 = 0. \tag{5.10}$$

Simple manipulations yield

$$\lambda = \frac{\alpha \ \beta + 2 \ c_1 - 1}{2 \ \beta} \ , \quad \omega = \frac{1}{2}\sqrt{\frac{1 - 2 \ c_1 - \alpha \ \beta}{\beta}}.$$

Eq (5.8) with the use of $\lambda$, $\omega$ and $k$ results in

$$u(g) = \frac{\alpha \ \beta + 2 \ c_1 - 1}{2 \ \beta} \ sec^2\left(\sqrt{\frac{1 - 2 \ c_1 - \alpha \ \beta}{4 \ \beta}} \ g\right).$$

Using inverse transformations, the solution of (5.7) in terms of original variables can be expressed as

$$u(t, x, y) = \frac{\alpha \ \beta + 2 \ c_1 - 1}{2 \ \beta} \ sec^2\left(\sqrt{\frac{1 - 2 \ c_1 - \alpha \ \beta}{4 \ \beta}} \ (x - \beta \ y + t \ (\alpha \ \beta - 1))\right),$$

$$v(t, x, y) = \frac{1}{\beta}\left[c_1 - \frac{\alpha \ \beta + 2 \ c_1 - 1}{2 \ \beta} \ sec^2\left(\sqrt{\frac{1 - 2 \ c_1 - \alpha \ \beta}{4 \ \beta}} \ (x - \beta \ y + t \ (\alpha \ \beta - 1))\right)\right],$$

where

$$u = c_1 - \beta \ v$$

which constitute the exact solutions of Eq (5.1).

## 6 Lie symmetries and exact solutions of the modified Calogero-Bogoyavlenskii-schiff (MCBS) equation

The modified Calogero-Bogoyavlenskii-schiff (CBS) equation is regulated by

$$v_t + v_{xxy} + 4 v^2 v_y + 4 v_x \partial_x^{-1} (v v_y) = 0, \tag{6.1}$$

which can be expressed into system of two equations

$$v_t + v_{xxy} + 4 v^2 v_y + 4 v_x u = 0,$$

$$\tag{6.2}$$

$$u_x = v v_y,$$

where $u = \partial_x^{-1} (v v_y)$ *and* $\partial^{-1}$ denotes the integral with respect to the subscript.

Using Eq (6.2), the following overdetermined linear set of PDEs are obtained

$$\xi_u^1 = 0, \qquad \xi_v^1 = 0, \qquad \xi_x^1 = 0, \qquad \xi_y^1 = 0, \qquad \xi_{tt}^1 = 0, \qquad \xi_u^2 = 0,$$

$$\xi_v^2 = 0, \qquad \xi_y^2 = 0, \qquad \xi_{tx}^2 = 0, \qquad \xi_{xx}^2 = 0, \qquad \xi_t^3 = 0, \qquad \xi_u^3 = 0,$$

$$\xi_v^3 = 0, \qquad \xi_x^3 = 0, \qquad \xi_y^3 = -2 \xi_x^2 + \xi_t^1, \qquad \eta^1 = \xi_x^2 u - u \xi_t^1 + \frac{1}{4} \xi_t^2, \qquad \eta^2 = -v \xi_x^2.$$

The following Lie point symmetries can be generated by solving determining equations with one component equal to one and the remaining equal to zero

$$X_1 = \frac{\partial}{\partial t} + \frac{\partial}{\partial x}, \qquad X_2 = \frac{\partial}{\partial x} + \frac{\partial}{\partial y}, \qquad X_3 = t \frac{\partial}{\partial t} + \frac{\partial}{\partial x} + y \frac{\partial}{\partial y} - u \frac{\partial}{\partial u}, \qquad X_4 = (x+1) \frac{\partial}{\partial x} - 2 y \frac{\partial}{\partial y} + u \frac{\partial}{\partial u} - v \frac{\partial}{\partial v}.$$

The combination of translational symmetries

$$X = X_1 + X_2 + \alpha X_3$$

or

$$X = \frac{\partial}{\partial t} + \frac{\partial}{\partial x} + \alpha \frac{\partial}{\partial y}.$$

The corresponding similarity transformations are

$$r = y - \alpha t, \quad s = x - t, \quad q = t, \quad u_y = u_r, \quad u_t = -\alpha u_r - u_s, \quad u_x = u_s.$$

Substituting above values in Eq (6.2), we obtain

$$-\alpha v_r - v_s + v_{ssr} + 4 v^2 v_r + 4 v_s u = 0,$$

$$\tag{6.3}$$

$$u_s - v v_r = 0.$$

System (6.3) admits the translational symmetries $X_1 = \frac{\partial}{\partial r}, \quad X_2 = \frac{\partial}{\partial s}$. Using combination of symmetries

$$X = \frac{\partial}{\partial r} + \beta \frac{\partial}{\partial s},$$

we find similarity transformations

$$g = s - \beta\, r, \quad h = r, \quad u_r = -\beta\, u_g, \quad u_s = u_g, \quad u_{rr} = \beta^2\, u_{gg}, \quad u_{ss} = u_{gg}.$$

Substituting above values in system (6.3), results in

$$-\beta\, v\, v_g - u_g = 0,$$

$$\frac{-\beta\, v^2}{2} - u + c_1 = 0.$$

(6.4)

From (6.4), we conclude

$$-\beta\, v_{ggg} - 6\, \beta\, v^2\, v_g + (\alpha\, \beta - 1 + 4\, c_1)\, v_g = 0.$$

Integrating w.r.t g and choosing constant of integration to zero gives rise to

$$-\beta\, v_{gg} - 2\, \beta\, v^3 + (\alpha\, \beta - 1 + 4\, c_1)\, v = 0.$$

(6.5)

Now using the sine-cosine approach [32–34], we obtain the explicit solutions of Eq (6.5). Suppose (6.5) has a solution such as

$$v(g) = \lambda\, cos^k(\,\omega\, g).$$

(6.6)

Substituting the values of $v$ from (6.6) in (6.5) yields

$$-2\,\beta\,\lambda^3\, (\,cos^k\,(\,\omega\, g))^3 + \lambda\, cos^k(\,\omega\, g)\,\beta\, k^2\,\omega^2 - \frac{\lambda\, cos^k\,(\,\omega\, g)\,\beta\, k^2\,\omega^2}{cos^2\,(\,\omega\, g)} + \lambda\, cos^k(\,\omega\, g)\,\alpha\,\beta$$
$$+ \frac{\lambda\, cos^k\,(\,\omega\, g)\,\beta\, k\,\omega^2}{cos^2\,(\,\omega\, g)} + 4\,\lambda\, cos^k(\,\omega\, g)\, c_1 - \lambda\, cos^k(\,\omega\, g) = 0.$$

(6.7)

Eq (6.7) is satisfied if $3k = k - 2$. Replacing $k = -1$ in (6.7) to obtain

$$-\frac{2\,\beta\,\lambda^3}{cos^3\,(\,\omega\, g)} + \frac{\lambda\,\beta\,\omega^2}{cos\,(\,\omega\, g)} - \frac{2\,\lambda\,\beta\,\omega^2}{cos^3\,(\,\omega\, g)} + \frac{\lambda\,\alpha\,\beta}{cos\,(\,\omega\, g)} + \frac{4\,\lambda\, c_1}{cos\,(\,\omega\, g)} - \frac{\lambda}{cos\,(\,\omega\, g)} = 0.$$

Comparing the coefficients of powers of $\dfrac{1}{cos(\,\omega\, g)}$, we have

$$-2\,\beta\,\lambda^3 - 2\,\lambda\,\beta\,\omega^2 = 0,$$

$$\lambda\,\beta\,\omega^2 + \lambda\,\alpha\,\beta + 4\,\lambda\, c_1 - \lambda = 0.$$

(6.8)

Solving (6.8) gives

$$\lambda = \sqrt{\frac{\alpha\,\beta + 4\, c_1 - 1}{\beta}}, \quad \omega = \sqrt{\frac{-\alpha\,\beta - 4\, c_1 + 1}{\beta}}\, g.$$

Using values of $\lambda$, $\omega$ and $k$ in (6.6) results in

$$v(g) = \sqrt{\frac{\alpha\,\beta + 4\,c_1 - 1}{\beta}}\ sec\left(\sqrt{\frac{-\alpha\,\beta - 4\,c_1 + 1}{\beta}}\,g\right),$$

$$u(g) = c_1 - \frac{\alpha\,\beta + 4\,c_1 - 1}{2}\ sec^2\left(\sqrt{\frac{-\alpha\,\beta - 4\,c_1 + 1}{\beta}}\,g\right).$$

The solution of (6.5) in terms of original variables can be expressed as

$$v(t,x,y) = \sqrt{\frac{\alpha\,\beta + 4\,c_1 - 1}{\beta}}\ sec\left(\sqrt{\frac{-\alpha\,\beta - 4\,c_1 + 1}{\beta}}\,( x - \beta\,y + t(\alpha\,\beta - 1))\right),$$

$$u(t,x,y) = c_1 - \frac{\alpha\,\beta + 4\,c_1 - 1}{2}\ sec\left(\sqrt{\frac{-\alpha\,\beta - 4\,c_1 + 1}{\beta}}\,(x - \beta\,y + t\,(\,\alpha\,\beta - 1))\right)^2. \tag{6.9}$$

Eq (6.9) constitute the exact solution of the modified Calogero-Bogoyavlenskii-schiff (CBS) Eq (6.2).

## 7 Lie symmetries and exact solutions of the modified KdV-CBS equation

The modified KdV-CBS equation is expressed as

$$u_t - 4\,u^2\,u_y - 2\,u_x\,\partial^{-1}(\,u^2)_y + u_{xxy} - 6\,u^2\,u_x + u_{xxx} = 0. \tag{7.1}$$

Eq (7.1) can be re-written as

$$u_t - 4\,u^2\,u_y - 2\,u_x\,v + u_{xxy} - 6\,u^2\,u_x + u_{xxx} = 0,$$

$$v_x - 2uu_y = 0, \tag{7.2}$$

where $v = \partial_x^{-1}(u^2)_y$ *and* $\partial^{-1}$ denotes the integral with respect to the subscripts.

Using Eq (7.2), we obtain the following set of overdetermined linear PDEs

$$\xi_u^1 = 0, \qquad \xi_v^1 = 0, \qquad \xi_x^1 = 0, \qquad \xi_y^1 = 0, \qquad \xi_{tt}^1 = 0, \qquad \xi_u^2 = 0,$$

$$\xi_v^2 = 0, \qquad \xi_x^2 = -\frac{1}{3}\,\xi_y^2 + \frac{1}{3}\,\xi_t^1, \qquad \xi_{ty}^2 = 0, \qquad \xi_{yy}^2 = 0, \qquad \xi_t^3 = 0,$$

$$\xi_u^3 = 0, \qquad \xi_v^3 = 0, \qquad \xi_x^3 = 0, \qquad \xi_y^3 = \frac{1}{3}\,\xi_t^1 + \frac{2}{3}\,\xi_y^2, \qquad \eta^1 = \frac{1}{3}\,u\,(\,\xi_y^2 - \xi_t^1),$$

$$\eta^2 = -\frac{1}{3}\,(\,3\,u^2 + v)\,\xi_y^2 - \frac{2}{3}\,v\,\xi_t^1 - \frac{1}{2}\,\xi_t^2. \tag{7.3}$$

We get the following Lie point symmetries

$$X_1 = \frac{\partial}{\partial t} + \frac{\partial}{\partial x}, \qquad X_2 = \frac{\partial}{\partial x} + \frac{\partial}{\partial y}, \qquad X_3 = t\,\frac{\partial}{\partial t} + (1 + \frac{1}{3}\,x)\,\frac{\partial}{\partial x} + \frac{1}{3}\,y\,\frac{\partial}{\partial y} - \frac{1}{3}\,u\,\frac{\partial}{\partial u} - \frac{2}{3}\,v\,\frac{\partial}{\partial v},$$

$$X_4 = (1 - \frac{1}{3}\,x + y)\,\frac{\partial}{\partial x} + \frac{2}{3}\,y\,\frac{\partial}{\partial y} + \frac{1}{3}\,u\,\frac{\partial}{\partial u} - (\frac{1}{3}\,v + u^2)\frac{\partial}{\partial v}.$$

The combination of translational symmetries

$$X = \frac{\partial}{\partial t} + \frac{\partial}{\partial x} + \alpha\,\frac{\partial}{\partial y}$$

rovide the similarity transformations

$$r = y - \alpha t, \quad s = x - t, \quad q = t, \quad u_y = u_r, \quad u_t = -\alpha\,u_r - u_s, \quad u_x = u_s. \tag{7.4}$$

Using change of variables (7.4), system (7.2) transforms to

$$-\alpha\,u_r - u_s + u_{ssr} - 4\,u^2\,u_r - 2\,u_s\,v - 6\,u^2\,u_s + u_{sss} = 0,$$

$$\tag{7.5}$$

$$v_s - 2\,u\,u_r = 0.$$

Eq (7.5) admits the Lie point symmetries $X_1 = \frac{\partial}{\partial r}$, $X_2 = \frac{\partial}{\partial s}$. We use the combination of symmetries

$$X = \frac{\partial}{\partial r} + \beta\,\frac{\partial}{\partial s},$$

to find the canonical variables

$$g = s - \beta\,r, \quad h = r, \quad u_r = -\beta\,u_g, \quad u_s = u_g, \quad u_{rr} = \beta^2\,u_{gg}, \quad u_{ss} = u_{gg}. \tag{7.6}$$

(7.5) with the help of similarity transformations and then integration w.r.t g, we find

$$v = c_1 - \beta\,u^2,$$

$$\tag{7.7}$$

$$(\alpha\,\beta - 1)\,u_g + 4\,\beta u^2\,u_g - 2\,u_g\,v - \beta\,u_{ggg} - 6\,u^2\,u_g + u_{ggg} = 0.$$

The solution of above system gives the following equation:

$$(1 - \beta)\,u_{ggg} + (\alpha\,\beta - 1 - 2\,c_1)\,u_g + 6\,\beta\,u^2\,u_g - 6\,u^2\,u_g = 0.$$

Integration w.r.t g gives rise to

$$(1 - \beta)\,u_{gg} + (\alpha\,\beta - 1 - 2\,c_1)\,u + 2\,(\beta - 1)\,u^3 = 0. \tag{7.8}$$

Now, using the sine-cosine approach [32–34], we obtain the explicit solutions of Eq (7.8). Suppose (7.8) has a solution such as

$$u(g) = \lambda\,cos^k\,(\,\omega\,g). \tag{7.9}$$

Substituting $u$ from (7.9) in (7.8) yields

$$2 \beta \lambda^3 \left( cos^k \left( \omega g \right) \right)^3 - 2 \lambda^3 \left( cos^k \left( \omega g \right) \right)^3 - \lambda cos^k \left( \omega g \right) k^2 \omega^2 + \lambda cos^k \left( \omega g \right) \beta k^2 \omega^2 - \frac{\lambda cos^k \left( \omega g \right) \beta k^2 \omega^2}{cos^2 \left( \omega g \right)}$$

$$+ \frac{\lambda cos^k \left( \omega g \right) k^2 \omega^2}{cos^2 \left( \omega g \right)} - \frac{\lambda cos^k \left( \omega g \right) k \omega^2}{cos^2 \left( \omega g \right)} + \lambda cos^k \left( \omega g \right) \alpha \beta - \lambda cos^k \left( \omega g \right)(2 c_1 + 1) + \frac{\lambda cos^k \left( \omega g \right) \beta k \omega^2}{cos^2 \left( \omega g \right)} = 0.$$

(7.10)

Eq (7.10) is satisfied if $2k = -2$. Substituting $k = -1$ in (7.10), we obtain

$$\frac{2 \beta \lambda^3}{cos^3 \left( \omega g \right)} + \frac{\lambda \beta \omega^2}{cos \left( \omega g \right)} - \frac{2 \lambda \beta \omega^2}{cos^3 \left( \omega g \right)} - \frac{2 \lambda c_1}{cos \left( \omega g \right)} + \frac{\lambda \alpha \beta}{cos \left( \omega g \right)} - \frac{\lambda}{cos \left( \omega g \right)} - \frac{\lambda \omega^2}{cos \left( \omega g \right)} + \frac{2 \lambda \omega^2}{cos^3 \left( \omega g \right)} - \frac{2 \lambda^3}{cos^3 \left( \omega g \right)} = 0.$$

Comparing the coefficients of powers of $\frac{1}{cos( \omega g)}$, we arrive at

$$\beta \lambda^2 - \lambda^2 - \beta \omega^2 + \omega^2 = 0,$$

(7.11)

$$\beta \omega^2 + \alpha \beta - \omega^2 - 2 c_1 - 1 = 0.$$

Solving (7.11) gives

$$\lambda = \sqrt{\frac{\alpha \beta - 2 c_1 - 1}{1 - \beta}}, \quad \omega = \sqrt{\frac{\alpha \beta - 2 c_1 - 1}{1 - \beta}} \, g.$$

Using the values of $\lambda$, $\omega$ and $k$ in (7.9) results in

$$u(g) = \sqrt{\frac{\alpha \beta - 2 c_1 - 1}{1 - \beta}} \, sec\left( \sqrt{\frac{\alpha \beta - 2 c_1 - 1}{1 - \beta}} \, g \right),$$

$$v(g) = c_1 - \beta \left( \frac{\alpha \beta - 2 c_1 - 1}{1 - \beta} \right) sec^2 \left( \sqrt{\frac{\alpha \beta - 2 c_1 - 1}{1 - \beta}} \, g \right).$$

Thus,

$$u(t, x, y) = \sqrt{\frac{\alpha \beta - 2 c_1 - 1}{1 - \beta}} \, sec\left( \sqrt{\frac{\alpha \beta - 2 c_1 - 1}{1 - \beta}} \, (x - \beta y + t \left( \alpha \beta - 1 \right)) \right),$$

$$v(t, x, y) = c_1 - \beta \left( \frac{\alpha \beta - 2 c_1 - 1}{1 - \beta} \right) sec^2 \left( \sqrt{\frac{\alpha \beta - 2 c_1 - 1}{1 - \beta}} \, (x - \beta y + t \left( \alpha \beta - 1 \right)) \right),$$

which constitute the exact solutions of the modified KdV-CBS equation (7.2).

## 8 Conclusion

The exact solutions of the $(1 + 1)$- dimensional integro-differential Ito, the first integro-differential KP hierarchy, the Calogero-Bogoyavlenskii-Schiff (CBS), the modified Calogero-Bogoyavlenskii-Schiff (CBS) and the modified KdV-CBS equations have been successfully established by utilizing the similarity transformations and the sine-cosine method. In the evaluation of exact solutions, the reduction in the number of independent variables via similarity variables and the use of inverse similarity transformations have been made. By substituting back, it has been checked that the acquired solutions satisfy the prescribed equations.

Furthermore, we have done the Lie symmetry analysis to investigate these solutions. Thus, the proposed approach is more efficient, reliable, and concise by means of computational complexity. It can provide more exact solutions as compared to the other methods that exist in the literature. The obtained solutions are new and innovative to existing ones and therefore, more appropriate to understand. In fact, the proposed method is readily applicable to a large variety of nonlinear evolution equations which frequently appear in mathematical physics and nonlinear sciences.

## Acknowledgments

The author expresses gratitude to the referees for their constructive criticism, which helped to strengthen the paper's substance.

## Author Contributions

**Conceptualization:** Amina Amin, Imran Naeem.

**Data curation:** Amina Amin.

**Formal analysis:** Amina Amin, Adnan Khan.

**Funding acquisition:** Amina Amin.

**Investigation:** Amina Amin.

**Methodology:** Amina Amin, Imran Naeem.

**Project administration:** Imran Naeem.

**Software:** Imran Naeem.

**Supervision:** Imran Naeem, Adnan Khan.

**Validation:** Amina Amin.

**Visualization:** Amina Amin, Imran Naeem.

**Writing – original draft:** Amina Amin.

**Writing – review & editing:** Amina Amin.

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
