## [Decision Letter · Decision Letter 0]

28 Nov 2022

PONE-D-22-25357Exact solutions of (1 + 1)-dimensional integro-differential Ito, KP Hierarchy, CBS, MCBS and Modified KdV-CBS EquationsPLOS ONE

Dear Dr. Amin,

Thank you for submitting your manuscript to PLOS ONE. After careful consideration, we feel that it has merit but does not fully meet PLOS ONE’s publication criteria as it currently stands. Therefore, we invite you to submit a revised version of the manuscript that addresses the points raised during the review process.

We look forward to receiving your revised manuscript.

Kind regards,

Shou-Fu Tian

Academic Editor

PLOS ONE

Journal Requirements:

Reviewers' comments:

Reviewer's Responses to Questions

**Comments to the Author**

1. Is the manuscript technically sound, and do the data support the conclusions?

Reviewer #1: Yes

Reviewer #2: Yes

2. Has the statistical analysis been performed appropriately and rigorously? 

Reviewer #1: Yes

Reviewer #2: Yes

3. Have the authors made all data underlying the findings in their manuscript fully available?

Reviewer #1: Yes

Reviewer #2: Yes

4. Is the manuscript presented in an intelligible fashion and written in standard English?

Reviewer #1: No

Reviewer #2: Yes

5. Review Comments to the Author

Reviewer #1: The (1 + 1)-dimensional integro-differential Ito, the first integro-differential KP hierarchy, the CBS, the modified CBS and the modified KdV-CBS equations are studied by the similarity reductions and sine-cosine methods. Some exact solutions of these equations are constructed by the similarity reductions and sine-cosine methods. The method is interesting for studying the nonlinear PDEs. The manuscript possesses lots of typos. I list some examples:

1. The contents of last five lines in page 3 should be modified.

2. P4 `` We Separate” should be `` we Separate”. ``After computing the values of alpha and beta, put in the main equation, we obtain the solution.” should be modified.

3. P16. `` modified Kdv-CBS equation” should be `` modified KdV-CBS equation”. ``the Similarity transformations” should be ``the similarity transformations”.

4. P17. ``Now, Using the sine-cosine” should be ``Now, using the sine-cosine”. ``Substituting k = -1 in (7.10), to obtain” should be modified.

5. P18. ``modified Kdv-CBS equation (7:2).” Should be ``modified KdV-CBS equation (7:2).” ``integro-differential (Ito),” should be ``integro-differential Ito,”.

English of whole paper should be checked for grammar.

In this paper, the Lie point symmetry method is applied to the nonlinear systems. The symmetry reductions related by the nonlocal symmetry are studied in ``Commun Nonlinear Sci Numer Simulat 42 (2017) 456; Nonlinear Dyn 86 (2016) 1855; Eur. Phys. J. Plus 131 (2016) 441”. The authors can add some discussions in this field.

The manuscript will be recommended for publication in the journal after revisions.

Reviewer #2: The present manuscript mainly studies “Exact solutions of (1 + 1)-dimensional integro-differential Ito, KP Hierarchy, CBS, MCBS and Modified KdV-CBS Equations”. Some basic definitions, important relations, and the fundamental theorem of the sine-cosine method are first presented. Then, Lie symmetries and exact solutions of (1 + 1)-dimensional integro-differential Ito equations, the first integro-differential KP hierarchy, the Calogero-Bogoyavlenskii-Schiff (CBS), the modified Calogero-Bogoyavlenskii-Schiff (CBS), and the modified KdV-CBS equations are constructed. Finally, the sine-cosine method is utilized to determine the exact solutions.

This manuscript is well written and scientifically sounds good, actual and important to the field. Its contents are interesting and helpful for wide audience. Moreover, the results represented in this manuscript enrich the dynamics of physical structures in nonlinear science. I believe that the results of this paper are correct.

The authors have stated some important methods which can be used to compute solutions of nonlinear PDEs in introduction. If the authors add the following content to introduction to discuss the latest progress of Dbar-steepest descent method in the section of INTRODUCTION, this paper will be more excellent.

“Recently, Li, Tian, Yang and Fan have done some interesting work in deriving the solutions of Wadati-Konno-Ichikawa equation and complex short pulse equation with the help of Dbar-steepest descent method. They solved the long-time asymptotic behavior of the solutions of these equations, and proved the soliton resolution conjecture and the asymptotic stability of solutions of these equations.(See: On the soliton resolution and the asymptotic stability of N-soliton solution for the Wadati-Konno-Ichikawa equation with finite density initial data in space-time solitonic regions，Adv. Math.，409 (2022) 108639; Soliton Resolution for the Wadati–Konno–Ichikawa Equation with Weighted Sobolev Initial Data，Ann. Henri Poincaré，23 (2022) 2611-2655; Soliton resolution for the complex short pulse equation with weighted Sobolev initial data in space-time solitonic regions，J. Differ. Equ. 329 (2022) 31–88.)”

Authors should carefully discuss these references in their introduction. I recommend the manuscript to be published after the minor modification.

6. PLOS authors have the option to publish the peer review history of their article (what does this mean?). If published, this will include your full peer review and any attached files.

Reviewer #1: No

Reviewer #2: No

---

## [Author Response · Author response to Decision Letter 0]

15 Feb 2023

Thank You so much for guiding and helping us to improve our article.

---

## [Decision Letter · Decision Letter 1]

13 Mar 2023

Exact solutions of (1 + 1)-dimensional integro-differential Ito, KP Hierarchy, CBS, MCBS and Modified KdV-CBS Equations

PONE-D-22-25357R1

Dear Dr. Amin,

We’re pleased to inform you that your manuscript has been judged scientifically suitable for publication and will be formally accepted for publication once it meets all outstanding technical requirements.

Kind regards,

Shou-Fu Tian

Academic Editor

PLOS ONE

Additional Editor Comments (optional):

Reviewers' comments:

Reviewer's Responses to Questions

**Comments to the Author**

1. If the authors have adequately addressed your comments raised in a previous round of review and you feel that this manuscript is now acceptable for publication, you may indicate that here to bypass the “Comments to the Author” section, enter your conflict of interest statement in the “Confidential to Editor” section, and submit your "Accept" recommendation.

Reviewer #1: All comments have been addressed

Reviewer #2: All comments have been addressed

2. Is the manuscript technically sound, and do the data support the conclusions?

Reviewer #1: Yes

Reviewer #2: Yes

3. Has the statistical analysis been performed appropriately and rigorously? 

Reviewer #1: Yes

Reviewer #2: Yes

4. Have the authors made all data underlying the findings in their manuscript fully available?

Reviewer #1: Yes

Reviewer #2: (No Response)

5. Is the manuscript presented in an intelligible fashion and written in standard English?

Reviewer #1: Yes

Reviewer #2: (No Response)

6. Review Comments to the Author

Reviewer #1: The authors have modified the manuscript according to the reviewers' comments. I recommend to publish the journal.

Reviewer #2: (No Response)

7. PLOS authors have the option to publish the peer review history of their article (what does this mean?). If published, this will include your full peer review and any attached files.

Reviewer #1: No

Reviewer #2: No

---

## [Editor Report · Acceptance letter]

20 Mar 2023

PONE-D-22-25357R1 

Exact solutions of (1 + 1)-dimensional integro-differential Ito, KP Hierarchy, CBS, MCBS and Modified KdV-CBS Equations 

Dear Dr. Amin:

I'm pleased to inform you that your manuscript has been deemed suitable for publication in PLOS ONE. Congratulations! Your manuscript is now with our production department. 

Kind regards, 

on behalf of

Dr. Shou-Fu Tian 

Academic Editor

PLOS ONE